# OpenReview forum: "HGMem: Hypergraph-based Working Memory to Improve Multi-step RAG for Long-Context Complex Relational Modeling"
_ICML.cc/2026/Conference — ICML 2026 regular_

### Official Review · Reviewer_KW4W · 2026-02-22

**Soundness:** 3
**Presentation:** 3
**Significance:** 2
**Originality:** 2
**Overall Recommendation:** 3
**Confidence:** 4

**Summary:**

This paper proposes HGMem, a novel hypergraph-based memory mechanism that transforms working memory into a dynamic and highly expressive knowledge structure.  In this approach, memory is modeled as a hypergraph where hyperedges act as memory points capable of connecting multiple entities to capture complex, high-order correlations. As the large language model interacts with external data through adaptive local and global retrieval, the memory progressively evolves via continuous update, insertion, and merging operations. Extensive evaluations on challenging global sense-making benchmarks demonstrate that HGMem consistently outperforms strong traditional and multi-step RAG baselines. Furthermore, the experiments reveal that HGMem powered by a smaller open-source model (Qwen2.5-32B-Instruct) can match or even exceed the performance of baseline systems relying on the more resource-intensive GPT-4o.

**Compliance With Llm Reviewing Policy:**

Affirmed.

**Key Questions For Authors:**

* What are the differences from and advantages over similar graph-based frameworks, e.g., [1-2].
* How sensitive is HGMEM's runtime performance to the quality, density, and accuracy of this initial offline graph?
* How does HGMEM mitigate error propagation if the LLM retrieves irrelevant or hallucinated information early in the multi-step process?


[1] From RAG to Memory: Non-Parametric Continual Learning for Large Language Models.

[2] G-reasoner: Foundation Models for Unified Reasoning over Graph-structured Knowledge.

**Limitations:**

Authors should discuss the limitations in the appendix.

**Strengths And Weaknesses:**

## Strengths

* The authors propose a technically sound hypergraph structure to address the limitation in LLM memory
* The authors evaluate HGMem against a robust set of baselines, including traditional RAG, graph-based RAG, and recent multi-step memory RAGs.
* The paper is well-structured and good writing

## Weaknesses

* The main contribution of this paper lies in the hypergraph-based memory structure, but its differences from and advantages over similar graph-based frameworks are not clearly articulated.
* Lack of baselines.
* The framework relies on an offline preprocessing stage using GPT-4o and LightRAG's tool to build the initial graph. This means HGMem's runtime effectiveness is fundamentally bottlenecked by the accuracy and coverage of this initial extraction.

---

> ### Author Rebuttal · Authors · 2026-03-30
>
> We sincerely thanks for your efforts in reviewing our work.
> # To W1&Q1
> Your mentioned [1,2] only constructs offline query-agnostic graphs to statically store primitive evidence for subsequent retrieval, while HGMem's core mechanism is intrinsically an online query-specific working memory that dynamically builds high-order correlations from progressively accumulated primitive evidence.
> ## Fundamental Methodological Differences:
> As classified in our Related Work, most existing graph-based frameworks build query-agnostic offline indexes that statically store information, which is intrinsically a static storage once being built, thus not specific to any query. In contrast, following the taxonomy in [3,4], working memory is usually query-specific, online built and dynamically maintained during resolving queries, temporarily storing accumulated information around a focal problem. In this sense, HGMem is intrinsically a query-specific working memory, operationalizing the concept of **dynamic memory evolving** rather than a static storage, as recognized by Reviewer vXY7. The initial offline graph merely serves as an auxiliary searching index, not the working memory itself.
> ## Advantages:
> The above differences just yield two core advantages:
> 1. HGMem dynamically evolves to gather highly pertinent, query-specific information, whereas the static graphs in [1,2] cannot.
> 2. While [1,2] statically store primitive evidence, HGMem explicitly **evolves** to form high-order correlations during online query resolving, enabling superior complex relational modeling, as evidenced by our experiments and analysis.
> # To W2
> Except for the baselines in the main results (Table 1), we also compare with several closely related strong methods published very recently in Appendix E.1 (KnowTrace, A-Mem, PropRAG), which further validates our HGMem's advantages over these recent strong methods. Besides, **it is noteworthy to clarify that your mentioned work [1] is well known as HippoRAG v2, which has been compared in the main results (Table 1).** Regarding [2] (G-reasoner), we find that its official repository currently does not provide open-source implementation, preventing a direct quantitative comparison.
>
>
> Meanwhile, we notice that, as the S2 in your reviews, you have appreciated that we compared “a robust set of baselines," which is contradictory to this weakness. So we kindly ask if your mentioned "lack of baselines" is just a misunderstanding. Or could you provide your intended baselines? We are fully open to include additional open-source baselines in the final revision.
>
> # To W3&Q2
> In fact, almost all graph-rag methods are largely dependent on offline graph construction. However, beyond a static offline graph, our HGMem can also dynamically evolve its hypergraph-based working memory during online query resolving. As elaborated in our response to your W1&Q1, this query-specific online evolving of working memory allows HGMem to continuously update and restructure information, thereby compensating for the omissions or inaccuracies of the initial static graph during runtime, which directly addresses your concern.
>
> Moreover, we add new experiments to test HGMem's performance with two variants of initial graphs of different densities or qualities. **You can refer to the results and analysis in our response to the W1&Q1 from Reviewer vXY7.**
> The results clearly indicate that although the initial graph affects HGMem’s runtime effectiveness to some extent, HGMem still mostly maintains its performance and significantly outperforms other baselines with initial graphs of varying quality/density.
>
> # To Q3
> HGMem mitigates early error propagation through continuous topological awareness and strict grounding in original source texts. If irrelevant or hallucinated information is retrieved early, the system recovers through two primary mechanisms:
> 1. During subsequent Local Investigation and Global Exploration (detailed in Section 3.4), the LLM actively scans the neighborhood of the vertices currently held in working memory. If contradictory or corrective information is discovered within this search scope, the LLM is prompted to adaptively overwrite or synthesize the context using the Update and Merge operations.
> 2. Every node/edge in the search space and every vertex/hyperedge in the working memory retains strict pointers to their original text chunks from the source document. Because the LLM always has access to golden textual references during memory updates, it can verify claims against the raw text, effectively recovering from potential intermediate hallucinations.
>
> [1] From RAG to Memory: Non-Parametric Continual Learning for Large Language Models
>
> [2] G-reasoner: Foundation Models for Unified Reasoning over Graph-structured Knowledge
>
> [3] Wu et al. From human memory to ai memory: A survey on memory mechanisms in the era of llms
>
> [4] Rethinking Memory in AI: Taxonomy, Operations, Topics, and Future Directions

---

> > ### Author Rebuttal · Reviewer_KW4W · 2026-04-05
> >
> > Thanks. Lacking of baselines is still my major concern. G-reasoner's previous work GFM-RAG has provided the complete implementation in [1], and also some search-enhanced methods are not included, e.g., search-r1.
> >
> > [1] https://github.com/RManLuo/gfm-rag

---

> > > ### Author Response · Authors · 2026-04-08
> > >
> > > **In the homepage and issuse page of your provided code repository[1], at the time being, it clearly notifies that the implementation of G-Reasoner is not currently available.** Therefore, as you requested, we can only implement the opensource code of GFM-RAG and Search-R1. Utilizing Qwen2.5-32B-Instruct as the backbone LLM, we compare GFM-RAG and Search-R1 with our HGMem. The experimental results are detailed below:
> > > | Methods/Accuracy | NarrativeQA(%)| Nocha(%) | Prelude(%) |
> > > | :--- | :--- | :--- | :--- |
> > > | HGMem | 	51.00 | 70.63 | 62.22 |
> > > | Search-R1 | 48.00 | 58.73 | 51.11 |
> > > | GFM-RAG | 34.00 | 61.11 | 55.56 |
> > >
> > > As demonstrated, HGMem consistently outperforms both Search-R1 and GFM-RAG across all benchmarks substantially. For GFM-RAG, as we categorized in our initial rebuttal, it just relies on a static query-agnostic offline index. When given a specific query, the query-agnostic nature of GFM-RAG impedes its performance on complex relational reasoning.
> > >
> > > Furthermore, it should also be noted that **DeepRAG, a core baseline thoroughly evaluated in our original manuscript, has been empirically shown to outperform Search-R1** under controlled model capacities and evaluation protocols [2]. Consequently, our initial evaluation suite already encompassed a highly competitive and representative paradigm, further corroborating the robust empirical standing of HGMem.
> > >
> > > Given these supplementary results and the fact that your other concerns have been fully resolved, would you consider raising your score?
> > >
> > > [1]https://github.com/RManLuo/gfm-rag/issues/32
> > >
> > > [2] DeepRAG: Thinking to Retrieve Step by Step for Large Language Models. (https://openreview.net/pdf?id=VI2YaggHIF)

---

### Official Review · Reviewer_jvXa · 2026-03-07

**Soundness:** 3
**Presentation:** 3
**Significance:** 3
**Originality:** 3
**Overall Recommendation:** 4
**Confidence:** 4

**Summary:**

This paper introduces HGMem, a hypergraph-based working memory mechanism designed to improve multi-step Retrieval-Augmented Generation (RAG) for complex tasks that require reasoning over long contexts. The key idea is to structure the evolving memory as a hypergraph, where vertices represent entities and hyperedges capture high-order relationships among multiple entities. Experiments on generative sense-making QA and long narrative understanding benchmarks show that HGMem outperforms both traditional and multi-step RAG baselines.

**Compliance With Llm Reviewing Policy:**

Affirmed.

**Key Questions For Authors:**

See the weaknesses.

**Limitations:**

See the weaknesses.

**Strengths And Weaknesses:**

Strengths:

S1: HGMem models working memory as a hypergraph, which can naturally represent high-order relationships among multiple entities . This goes beyond simple pairwise edges used in prior graph-based RAG methods and enables more complex reasoning, as demonstrated in the case study where the model correctly infers causal chains.

S2: The paper goes beyond aggregate metrics by distinguishing primitive  and sense-making (reasoning-intensive) queries. This analysis  reveals that HGMem's advantage comes from building higher-order correlations for complex questions, while it does not harm performance on simple ones. The step-wise evaluation shows that performance peaks at step 3, providing practical guidance on iteration depth.

Weaknesses:

W1: The paper does not report end-to-end latency or cost for the offline graph construction, which could be substantial for very large documents. This complexity may hinder adoption in resource-constrained settings.

W2: For the Longbench V2 tasks, the paper uses comprehensiveness and diversity scores assigned by GPT-4o as a judge. Although the authors provide detailed rubrics, such subjective metrics are inherently noisy and may favor responses that align with GPT-4o's own style. The paper would be stronger if it included human evaluation or objective metrics alongside the LLM-based scores to validate the trends.

---

> ### Author Rebuttal · Authors · 2026-03-30
>
> We sincerely thank you for the efforts taken in reviewing our work. We hope the following responses would help address your concerns.
> # To W1
> We agree that the end-to-end latency and cost for LLM-based offline graph construction is affected by the scale of documents. However, we respectfully clarify that **the specific method of offline graph construction is orthogonal to the core contribution of HGMem**. The cost of offline graph construction is highly dependent on the chosen preprocessing approach. In addition, HGMem can make use of any form of graph-structured index, which is not necessarily built using LLMs.
> Consequently, in resource-constrained environments, practitioners can substitute computationally expensive LLMs with lightweight, traditional extraction tools without compromising the applicability of our framework.
> To further validate our above arguments, we carry out a new set of experiments as follows.
> - Exp1: HGMem with partially ablated LLM-generated offline graph (ablate 50%)
> - Exp2: Graph constructed by traditional tools (Stanford OpenIE)
>
> |Methods|NarrativeQA Acc(%)|Nocha Acc(%)|Prelude Acc(%)|
> |-|-|-|-|
> |Exp 1| - | - | - |
> |HGMem| 48.00 | 68.25 |57.97|
> |LightRAG| 33.00 | 57.14 | 54.81 |
> |DeepRAG| 42.00 | 65.08 | 49.63 |
> |Exp 2| - | - | - |
> |HGMem| 50.00 | 66.67 | 59.26 |
> |LightRAG| 36.00 | 57.93 | 53.33 |
> |DeepRAG| 42.00 | 63.49 | 47.41 |
>
> These new results demonstrate the ***consistent performance advantage of our HGMem in resource-constrained scenarios where large-scale LLM-based preprocessing is not feasible.***
>
> Besides, in real-world industrial implementation, the cost of online multi-step RAG execution is usually the major concern as the offline constructed graph can be repeatedly exploited once built in advance. Since our core innovation lies in the dynamic evolution of working memory to form higher-order correlations, we focused our cost analysis on the **online execution phase**. As detailed in Appendix E.2, this analysis confirms that HGMem achieves superior reasoning performance while maintaining runtime costs on par with typical multi-step RAG baselines.
>
> Another reason for using LLM to preprocess raw documents into offline graphs is to make fair comparisons against existing graph-based RAG approaches like GraphRAG, LightRAG and HippoRAG v2, all of which necessarily involve using LLM to build the offline static query-agnostic graph.
>
> We hope the above discussion and the experimental results would resolve your concern regarding the LLM-based offline graph construction.
>
>
> # To W2
> Thanks for your constructive suggestions about adding human evaluation. However, before adding supplementary experiments, we find it necessary to clarify the critical misunderstanding about the concern that GPT-4o might favor its own output. **In all experiments (e.g. Table 1), performances of different methods are fairly compared under the same backbone LLM (GPT-4o or Qwen2.5)**. The real thing that matters is their **relative performances across diverse methods with the same backbone.** In other words, the performances should be compared across different baseline methods rather than across different backbone LLMs. Therefore, your concern does not affect the validity of our analysis and conclusion.
>
> Furthermore, we provide the supplementary human evaluation as follows. Specifically, we uniformly sample a total of 100 instances from Longbench V2 used in our original experiment. Then, following the selection-based evaluation used in [1,2], given a query and two responses from different methods with Qwen2.5-32B as the base model, a group of three PHD-level students are asked to select the better one in terms of the comprehensiveness and diversity of the responses (defined the same as) based on the corresponding paragraph. In this way, the win rates of each method can reflect human preference towards the responses from different methods. Due to the limited duration of the rebuttal period, we just ask them to compare our HGMem to the strongest multi-step RAG baseline DeepRAG. The win rates of the two methods in terms of comprehensiveness and diversity are as follows:
>
> | Methods | Comprehensiveness | Diversity |
> |----|----|----|
> |HGMem|78%|71%|
> | DeepRAG|22%|29%|
> | Sum |100%| 100%|
>
> These results further quantitatively strengthen the ***reliability of trends reported in our original paper***, which we will add into the revised version.
>
> We hope the above clarification and experiment results would resolve your misunderstanding about experimental setups and metrics.
>
> ### References:
>
> [1] From local to global: A graph RAG approach to query-focused summarization.
>
> [2] Lightrag: Simple and fast retrieval-augmented generation

---

> > ### Author Rebuttal · Reviewer_jvXa · 2026-04-05
> >
> > Thanks for your responses. No more questions.

---

> > > ### Author Response · Authors · 2026-04-06
> > >
> > > Thanks for your kind reply. We are encouraged that your concerns have been fully resolved. Given that there is no unresolved weakness, would you consider raising the score? If there is still something necessary for us to provide (e.g. more detailed clarification or new experiments), we are very happy and fully open to add that.

---

### Official Review · Reviewer_vXY7 · 2026-03-12

**Soundness:** 4
**Presentation:** 3
**Significance:** 4
**Originality:** 4
**Overall Recommendation:** 5
**Confidence:** 4

**Summary:**

This paper proposes HGMem, a novel hypergraph-based working memory mechanism for multi-step RAG. It tackles a key limitation of static memory in existing systems by modeling memory as a dynamic hypergraph, where memory units (hyperedges) can evolve through LLM-guided update, insertion, and merging operations. This allows for the progressive formation of high-order correlations among facts, aiming to improve reasoning in long-context, complex relational tasks. The paper is well-supported by extensive experiments.

**Compliance With Llm Reviewing Policy:**

Affirmed.

**Final Justification:**

The paper is technically sound and presents a novel hypergraph-based framework for dynamic working memory in multi-step RAG. My main concerns regarding dependence on high-quality offline knowledge graphs and the paper’s positioning have been effectively addressed in the rebuttal through additional experiments and clarifications. This strengthens my confidence in the robustness and contribution of the work. I support an Accept recommendation.

**Key Questions For Authors:**

1. Would the authors agree that the most precise description of HGMem’s contribution is a **novel system architecture/prompting framework**? Furthermore, to address concerns about dependency on input quality, could the authors comment on the feasibility and potential results of a sensitivity analysis using a simpler method to construct the foundational knowledge graph?
2. Beyond storage and retrieval anchoring, does the current implementation use any **graph-theoretic algorithms** (e.g., community detection, centrality measures) that compute on the hypergraph’s topology to inform decisions? If not, would the authors frame the hypergraph’s current role primarily as a **powerful organizational representation**, with deeper algorithmic integration being a key direction for future work?
3. The analysis in Sec. 5.4 is excellent. To add rigor, could the authors report the **inter-annotator agreement** (e.g., Cohen’s Kappa) for the manual query categorization? Additionally, if efficiency data exists in the appendix, a brief discussion of the **performance-overhead trade-off** in the main text would provide readers with a more complete view for practical assessment.

**Limitations:**

yes

**Strengths And Weaknesses:**

**Strengths:**

- **Soundness**: The experimental design is robust. The baselines are comprehensive. The ablation studies are well-designed, directly validating core components like the merging operation.
- **Presentation**: The paper is clearly written and well-structured. The motivation is clear, and the methodology description is sufficiently detailed for reproducibility.
- **Significance**: The work addresses a practical and important problem—enhancing LLMs' ability for complex relational reasoning in long contexts. The proposed dynamic memory evolution framework offers a new design perspective for multi-step RAG systems.
- **Originality**: Introducing hypergraphs for working memory modeling is a novel idea. Operationalizing "dynamic memory evolution" as concrete operations on a hypergraph structure constitutes an original systems contribution.

**Weaknesses and Suggestions:**

- **Soundness**:
  1. The performance of HGMem is evaluated using a knowledge graph constructed offline with a powerful model (GPT-4o). While this establishes a strong upper bound, it leaves open the question of how much the observed gains depend on this high-quality input versus the HGMem framework itself. A discussion or supplementary analysis on the method's sensitivity to the quality of the foundational knowledge graph would strengthen the claim of the framework's inherent value and generalizability.
- **Presentation**:
  1. The paper’s innovation is best described as a novel **system architecture** that uses a hypergraph as a dynamic state representation, manipulated via LLM prompting. However, some phrasing (e.g., emphasis on “leveraging topological structure”) could lead readers to expect graph-theoretic algorithms as the primary reasoning engine. The paper would benefit from more explicitly framing HGMem as a **prompting framework and system design** in the abstract and introduction, clarifying that the LLM is the reasoning engine operating on the hypergraph state.
  2. In the current implementation, the hypergraph primarily serves as an expressive **storage representation** and a **scaffold for retrieval** (e.g., local anchors). The critical “merging” operation is based on LLM judgment, not on algorithmic analysis of the hypergraph’s topology. The authors could enhance the discussion by explicitly stating this focus on “hypergraph as a structured representation” and outlining the potential for future algorithmic exploitation of the topology.

---

> ### Author Rebuttal · Authors · 2026-03-30
>
> We sincerely thank the reviewer for the constructive and encouraging feedback. We address the weakness and question below.
> # To W1&Q1
> Before further discussion, we would first like to clarify two points:
> 1. In original experiments, as shown in Table 1, HGMem and the working memory-based baselines were compared under the exact same initial graph. Thus, the observed gains of HGMem benefit mainly from the framework itself.
> 2. HGMem can make use of any form of graph-structured index of different densities or quailities. Particularly, in resource-constrained scenarios, it is also feasible to apply a simpler offline graph construction method (e.g. LLM-free tools) to extract a lower-quality offline knowledge graph.
>
> To further validate HGMem’s generalizability and sensitivity to different graph qualities, we conducted additional experiments using Qwen2.5-32B-Instruct as HGMem's backbone LLM with the following variants of pre-built offline graph.
>
> - Variant 1: Partially ablated offline graph (randomly ablate 50% entities&relationships)
> - Variant 2: HGMem with the graph constructed by traditional LLM-free tools (Stanford OpenIE).
>
> The experimental results are as follows:
> |Methods|NarrativeQA Acc(%)|Nocha Acc(%)|Prelude Acc(%)|
> |-|-|-|-|
> |Original| - | - | - |
> |HGMem| 51.00 | 70.63 | 62.22 |
> |LightRAG| 40.00 | 59.52 | 60.74|
> |DeepRAG| 44.00 | 66.40 | 51.11 |
> |Variant 1| - | - | - |
> |HGMem| 48.00 | 68.25 |57.97|
> |LightRAG| 33.00 | 57.14 | 54.81 |
> |DeepRAG| 42.00 | 65.08 | 49.63 |
> |Variant 2| - | - | - |
> |HGMem| 50.00 | 66.67 | 59.26 |
> |LightRAG| 36.00 | 57.93 | 53.33 |
> |DeepRAG| 42.00 | 63.49 | 47.41 |
>
> The results show that although the quality of the offline graph would affect HGMem to some extent, our HGMem still consistently achieves a ***significant performance advantage*** when the offline graph is partially ablated or built by simpler LLM-free tools. Overall, it demonstrates that the majority of the ***observed gains are intrinsic to the HGMem framework itself***.
> Besides, it can also be seen that all of HGMem and the compared methods are affected by the density and quality of the initial graph to a similar extent, indicating ***HGMem's moderate sensitivity to the initial graph*** compared to other methods.
>
> # To W2&W3&Q2
> Thanks for your kind suggestion.
> We mostly agree with your opinion that HGMem is largely a system design where the LLM basically serves as the reasoning engine operating on the hypergraph-based working memory state. Nevertheless, it is also notable to mention that there are actually some mechanisms leveraging graphical topology structure, especially the mechanims elaborated in Section 3.4 Adaptive Memory-based Evidence Retrieval including **"local investigation" (Eq.6)** and **"global exploration" (Eq.7)**. As for your mentioned graph-theoretic algorithms (e.g., community detection, centrality measures), we respectfully acknowledge that it would be a valuable potential direction for integrating deeper algorithms in future work, which enables the LLM to be more aware of the hypergraph topology beyond current hypergraph-based storage representation and adaptive topological retrieval scaffold. Once accepted, we will more precisely refine relevant descriptions in the abstract and introduction in the camera-ready revision.
>
> # To Q3
> Thanks for your constructive advice. Here, we give more detailed statistics about the query categorization. Among the 120 sampled instances, we ask three PhD-level students to carry out manual categorization, where their agreement in terms of Fleiss’s Kappa is **0.88**. We promise to add the discussion of efficiency to the main text in the future version.

---

> > ### Author Rebuttal · Reviewer_vXY7 · 2026-04-01
> >
> > Thank you for the author's detailed response. I am satisfied with the reply, as the additional sensitivity experiments (low-quality knowledge graphs) effectively addressed the core concern about the method's reliance on high-quality input. At the same time, the authors clarified that HGMem is an innovative system architecture/prompting framework and committed to revising the main text, which makes the paper's contribution clearer and more accurate. The provided inter-annotator agreement also strongly supports the rigor of the analysis. Regarding the efficiency discussion, the authors have promised to include it in the final version—a minor but necessary refinement.
> >
> > I have decided to raise my rating to Accept. This is an original and well-argued work on dynamic working memory mechanisms.

---

> > > ### Author Response · Authors · 2026-04-03
> > >
> > > Thanks for your highly constructive review and raising the overall rating. We are very encouraged that our response and additional experiments effectively addressed your core concerns. As promised, we will carefully incorporate your excellent suggestions on efficiency discussion, system clarification, and sensitivity analysis into the final revision. We deeply appreciate your time and support.

---

### Decision · Program_Chairs · 2026-04-30

**Decision:**

Accept (regular)

**Comment:**

This paper proposes a hypergraph-based working memory mechanism for multi-step RAG and evaluates it on several long-context, reasoning-intensive benchmarks. In my view, the paper makes a solid contribution. The central idea is interesting, the experimental results are strong, and the rebuttal did a good job addressing the main technical concerns raised in review.

In particular, the authors provided additional evidence on sensitivity to the quality of the initial offline graph, clarified the role of the hypergraph in the current system, and added further support for the evaluation setup. These responses substantially strengthened the paper. The remaining concerns are mostly about framing and scope rather than correctness: the current implementation is better understood as a dynamic memory / systems design that uses a hypergraph as a structured working-memory representation, rather than as a graph-theoretic reasoning method in the algorithmic sense. I think that should be stated more clearly in the final version.

One reviewer continued to worry about baseline coverage, but the authors added further comparisons and the overall empirical case remains convincing. On balance, I find the paper technically sound, empirically well supported, and likely to be useful to the community working on RAG and long-context reasoning.